# OS-net: Orbitally Stable Neural Networks Conference Submissions

## Abstract

We introduce OS-net (Orbitally Stable neural NETworks), a new family of neural network architectures specifically designed for periodic dynamical data. OS-net is a special case of Neural Ordinary Differential Equations (NODEs) and takes full advantage of the adjoint method based backpropagation method. Utilizing ODE theory, we derive conditions on the network weights to ensure stability of the resulting dynamics. We demonstrate the efficacy of our approach by applying OS-net to discover the dynamics underlying the Rössler and Sprott's systems, two dynamical systems known for their period doubling attractors and chaotic behavior.

## 1 Introduction

The study of periodic orbits of systems of the form

$$\dot{\boldsymbol{x}} = f(\boldsymbol{x}), \ \boldsymbol{x}(0) = \boldsymbol{x}_0, \ \boldsymbol{x} \in \mathbf{U} \subset \mathbf{R}^n \tag{1}$$

is an important area of research within the field of nonlinear dynamics with applications in both the physical (astronomy, meteorology) and the nonphysical (economics, social psychology) sciences. In particular, periodic orbits play a significant role in chaos theory. In Devaney (2003), chaotic systems are defined as systems that are sensitive to initial conditions, are topologically transitive (meaning that any region of the phase space can be reached from any other region), and have dense periodic orbits. Notably, chaotic systems are constituted of infinitely many Unstable Periodic Orbits (UPOs) which essentially form a structured framework, or a "skeleton", for chaotic attractors. A periodic orbit is (orbitally) unstable if trajectories that start near the orbit do not remain close to it. Finding and stabilizing UPOs is an interesting and relevant research field with numerous applications such as the design of lasers Roy et al. (1992), the control of seizure activities Schiff et al. (1994) or the design of control systems for satellites Wiesel & Shelton (1983). An important tool when studying the stability of periodic orbits of a given system is the Poincaré or return map which allows one to study the dynamics of this system in a lower dimensional subspace. It is well-known that the stability of a periodic orbit containing a point $\boldsymbol{x}_0$ is inherently connected to the stability of $\boldsymbol{x}_0$ as a fixed point of the corresponding Poincaré map. However, explicitly computing Poincaré maps has been proven to be highly challenging and inefficient Teschl (2012). With the emergence of data-driven approaches, researchers in Bramburger & Kutz (2020) proposed a data-driven computation of Poincaré maps using the SINDy method Brunton et al. (2016). Subsequently, they leveraged this technique in to develop a method for stabilizing UPOs of chaotic systems Bramburger et al. (2021).

As a matter of fact, researchers have been increasingly exploring the intersection of machine learning and differential equations in recent years. For example, Partial Differential Equations (PDEs) inspired neural network architectures have been developed for image classification in Ruthotto & Haber (2019); Sun & Zhang (2020). On the other hand, data-driven-based PDE solvers were proposed in Sirignano & Spiliopoulos (2018) while machine learning has been effectively utilized to discover hidden dynamics from data in Raissi et al. (2019); Brunton et al. (2016); Schaeffer et al. (2017). One notable example of such intersectional work is Neural Ordinary Differential Equations (NODEs), which were introduced in Chen et al. (2018). NODEs are equivalent to continuous residual networks that can be viewed as discretized ODEs Haber & Ruthotto (2017). This innovative approach has led to several extensions that leverage well-established ODE theory and methods Dupont et al. (2019); Zhang & Zhao (2022); Ott et al. (2021); Zhuang et al. (2020); Yan et al. (2020); Haber & Ruthotto (2017) to develop more stable, computationally efficient, and generalizable architectures.

In the present work, we aim at learning dynamics obtained from chaotic systems with a shallow network featuring a single hidden layer, wherein the network's output serves as a solution to the dynamical system

$$\dot{\boldsymbol{x}} = \boldsymbol{W}_d^T \sigma(\boldsymbol{W}_e^T \boldsymbol{x} + \boldsymbol{b}_e), \quad \boldsymbol{x}(0) = \boldsymbol{x}_0. \tag{2}$$

where $\boldsymbol{W}_e$ are the input-to-hidden layer weights, $\boldsymbol{b}_e$ the corresponding bias term, $\boldsymbol{W}_d$ the hidden-to-output layer weights, and $\sigma$ the activation function of the hidden layer. The proposed network is a specific case of NODEs and fully utilizes the adjoint method-based Pontryagin (1987) weight update strategy introduced in Chen et al. (2018). Our primary objective is to establish sufficient conditions on the network parameters to ensure that the resulting dynamics are orbitally stable. We base our argument on the finding that the stability of Poincaré maps is equivalent to the stability of the first variational equation associated with the dynamical system under consideration Teschl (2012). We then build on the stability results of linear canonical systems presented in Krein (1983) to derive a new regularization strategy that depends on the matrix $\boldsymbol{J} = \boldsymbol{W}_e^T \boldsymbol{W}_d^T$ and not on the weight matrices taken independently. We name the constructed network OS-net for Orbitally Stable neural NETworks.

Since we are dealing with periodic data, the choice of activation function is critical. Indeed, popular activation functions such as the $\mathrm{sigmoid}$ or the $\mathrm{tanh}$ functions do not preserve periodicity outside the training region. A natural choice would be sinusoidal activations however these do not hold desired properties such as monotonicity. Furthermore, they perform poorly Parascandolo et al. (2017) on the training phase because the optimization can stagnate in a local minimum because of the oscillating nature of sinusoidal functions. In Ngom & Marin (2021), the authors constructed a Fourier neural network (i.e a neural network that mimics the Fourier Decomposition) Silvescu (1999); Zhumekenov et al. (2019) that uses a $\mathrm{sin}$ activation but had to enforce the periodicity in the loss function to ensure that periodicity is conserved outside of the training region. The activation functions $x + \frac{1}{a}\sin^2(ax)$ -called snake function with frequency $a$- and $x + \sin x$ were proposed in Ziyin et al. (2020) for periodic data and were proven to be particularly well suited to periodic data. As such, we use both these activation functions in this work.

This paper is organized as follows: in section2 we present the OS-net's architecture and the accompanying new regularization strategy. In section3 we showcase its performance on simulated data from the chaotic Rössler and Sprott systems and perform an ablation study to assess the contributions of the the different parts of OS-net.

## 2 BUILDING OS-NET

### 2.1 BACKGROUND

In this chapter, we recall the main results on the stability of periodic orbits of dynamical systems we will be using to build OS-net. We refer readers to the appendices for more details about orbits of dynamical systems.

We consider the system

$$\dot{\boldsymbol{x}} = f(\boldsymbol{x}), \quad \boldsymbol{x}(0) = \boldsymbol{x}_0. \tag{3}$$

and suppose it has a periodic solution $\phi(t, x_0)$ of period $T$. We denote $\gamma(x_0)$ a periodic orbit corresponding to $\phi(t, x_0)$. Stability of periodic orbits have been widely studied in the literature. It is, in particular well-known (Teschl, 2012, Chapter 12) that the stability of periodic orbits of Equation3 is linked to the stabiity of its First Variational (FV) problem

$$\dot{\boldsymbol{y}} = \boldsymbol{A}(t)\boldsymbol{y}, \quad \boldsymbol{A}(t) = d\left(f(\boldsymbol{x})\right)_{(\Phi(t, \boldsymbol{x}_0))} \quad \text{and} \quad \boldsymbol{A}(t + T) = \boldsymbol{A}(t). \tag{4}$$

which is obtained by taking the gradient of Equation3 with respect to $x$ at $\phi(t, x_0)$. As such, the first variational problem describes the dynamics of the state variable $y = d\left(\phi(t, t_0, \boldsymbol{x})\right)$ and is a linear system as the matrix $\boldsymbol{A}(t)$ does not depend on $y$.

To assess the stability of OS-net, we investigate the first variational equation associated with Equation2. OS-net's FV is given by

$$\dot{\boldsymbol{y}} = \boldsymbol{W}_d^T diag\left(\sigma'\left(\boldsymbol{W}_e^T \phi(t, \boldsymbol{x}_0) + \boldsymbol{b}_e\right)\right) \boldsymbol{W}_e^T \boldsymbol{y},$$

and if we make the change of variables $\boldsymbol{z} = \boldsymbol{W}_e^T \boldsymbol{y}$, this equation becomes

$$\dot{\boldsymbol{z}} = \boldsymbol{J}\boldsymbol{H}(t)\boldsymbol{z}, \tag{5}$$

where $J = W_e^T W_d^T$ and $H(t) = diag\left(\sigma'\left(W_e^T \phi(t, x_0) + b_e\right)\right)$ is periodic. This formulation can be seen as a generalization of linear canonical systems with periodic coefficients

$$\dot{y} = \lambda J_m H(t) y \tag{6}$$

where

$$J_m = \begin{pmatrix} 0 & I_m \\ -I_m & 0 \end{pmatrix}, I_m \text{ is the identity matrix of size } m,$$

$H$ is a periodic matrix-valued function and $\lambda \in \mathrm{R}$. Stability of such systems was extensively studied in Krein & Jakubovic (1983) and in particular in Krein (1983). We recall the main definitions and results from Krein (1983) and build upon these to derive stability conditions for OS-net. In particular, we give the definition of stability zones for Equation6 and provide the main stability results we will base our study on.

**Definition 1.** *A point $\lambda = \lambda_0$ $(-\infty < \lambda_0 < \infty)$ is called a $\lambda$-point of stability of Equation 6 if for $\lambda = \lambda_0$ all solutions of Equation 6 are bounded on the entire t-axis.*
*If, in addition, for $\lambda = \lambda_0$ all solutions for any equation*

$$\dot{y} = \lambda J_m H_1(t) y$$

*with a periodic symmetric. matrix valued function $H_1(t) = H_1(t+T)$ sufficiently close to $H(t)$ are bounded, then $\lambda_0$ is a $\lambda$-point of strong stability of Equation 6.*

The set of $\lambda$-point of strong stability of Equation 6 is an open-set that decomposes into a system of disjoint open intervals called $\lambda$-zones of stability of Equation6. If a zone of stability contains the point $\lambda = 0$ then it is called a central zone of stability.

**Definition 2.** *We say that Equation 6 is of positive type if*

$$H \in \mathrm{P}_n(T) = \{A(t) \text{ symmetric s.t } A(t) \geq 0 \ (0 \leq t \leq T) \text{ and } \int_0^T A(t)dt > 0\}.$$

$A(t) \geq 0$ means $\forall x \in \mathbf{R}^n, \langle A(t)x, \ x \rangle \geq 0$ and $\int_0^T A(t)dt > 0$ means $\int_0^T \langle H(t)x, \ x \rangle dt > 0$

**Definition 3.** *Let $A$ be a square matrix with non-negative elements. We denote by $\mathcal{M}(A)$ the least positive eigenvalue among its eigenvalues of largest modulus. Note that Perron's theorem (1907) guarantees the existence of $\mathcal{M}(A)$ Horn & Johnson (2012).*

We can now state the main result we will derive our regularization from:

**Theorem 1** (Krein (1983) section 7, criterion $I_n$). *A real $\lambda$ belongs to the central zone of stability of an Equation 6 of positive type, if*

$$|\lambda| < 2\mathcal{M}^{-1}(C)$$

*where $C = J_{m_a} \int_0^T H_a(t)$. If $K$ is a matrix, $K_a$ is the matrix obtained by replacing the elements of $K$ by their absolute values.*

The proof of this theorem is recalled in the appendices.

## 2.2 ARCHITECTURE AND STABILITY OF OS-NET

To base the stability of OS-net on stability theory for systems of type Equation6, we need the matrix-valued function $H(t)$ and the matrix $J$ in Equation 5 to be respectively of positive type and skew-symmetric. To ensure $H(t)$ is of positive type, it is sufficient to use activation functions that are increasing since they have positive derivatives and diagonal matrices with positive elements are of positive type. Fortunately, many common activation functions ($\tanh$ or $sigmoid$) have that property. In this paper, we use the strictly monotonic activation functions $x + \sin(x)$ and (the snake function) $x + \frac{1}{a}\sin^2(ax), \ a \in \mathbf{R}$ displayed in Figure1. These activation functions were proved to be able to learn and extrapolate periodic functions in Ziyin et al. (2020).

Let us now pay attention to the matrix $J = W_e^T W_d^T$. To ensure $J$ is skew-symmetric, we introduce the matrices $W \in \mathrm{M}_{n,2m}(\mathbf{R})$ and $K \in \mathrm{M}_{2m}(\mathbf{R})$ where $n$ is the input size (i.e the size of $x$) and $2m$ the number of nodes in the hidden layer. We then set $W_e^T = W, W_d^T = \Omega W^T$, and $\Omega = K - K^T$.

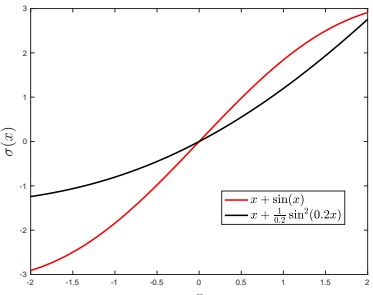 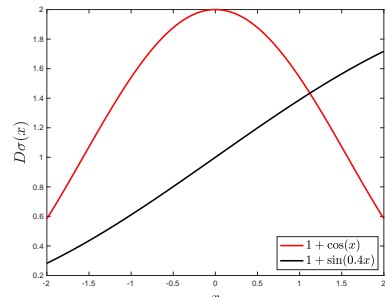

Figure 1: Left: Activation functions $x + \sin(x)$ and $x + \frac{1}{0.2}\sin^2(0.2x)$. Right: Derivatives of the activation functions

Note that the size of the hidden layer which is the size of $\boldsymbol{\Omega}$ needs to be even. Otherwise, $\boldsymbol{\Omega}$ would be a singular matrix. The elements of the matrices $\boldsymbol{W}$ and $\boldsymbol{K}$ are the hyperparameters of the network that will be optimized during training. Now, knowing that any real skew-symmetric matrix $\boldsymbol{J}$ is congruent to $\boldsymbol{J}_m$ Yakubovich & Starzhinskii (1975), there exists a real invertible matrix $\boldsymbol{S}$ such that

$$\boldsymbol{J}_m = \boldsymbol{S}^T \boldsymbol{J} \boldsymbol{S}$$

and Equation5 is equivalent to Equation6. In fact, let $\boldsymbol{u} = \boldsymbol{S}^T \boldsymbol{z}$ in Equation5, we obtain

$$\dot{\boldsymbol{u}} = \lambda \boldsymbol{J}_m \boldsymbol{S}^{-1} \boldsymbol{H}(t)(\boldsymbol{S}^{-1})^T \boldsymbol{u} = \lambda \boldsymbol{J}_m \tilde{\boldsymbol{H}}(t) \boldsymbol{u}$$

where $\tilde{\boldsymbol{H}}(t) = \boldsymbol{S}^{-1} \boldsymbol{H}(t)(\boldsymbol{S}^{-1})^T \in \mathrm{P}_n(T)$. We can now apply Theorem1 to OS-net and state that OS-net is stable if

$$1 < 2\mathcal{M}^{-1}\left( \boldsymbol{J}_a \int_0^T \boldsymbol{H}(t)dt \right). \tag{7}$$

Note that since $\boldsymbol{H}(t)$ is a diagonal matrix with positive elements, $\boldsymbol{H}_a(t) = \boldsymbol{H}(t)$. We can now prove the following result that will justify our regularization strategy:

**Corollary 1.1.** *Suppose the activation function $\sigma$ is strictly increasing with a uniformly bounded derivative. Then, OS-net is stable if*

$$||\boldsymbol{J}_a||_2 < \frac{2}{LT} \tag{8}$$

*where $L$ is the superior bound of the derivative of the activation function.*

*Proof.* Let $\mu$ be any eigenvalue of $\boldsymbol{J}_a \int_0^T \boldsymbol{H}(t)dt$ then $|\mu| \leq \left\| \boldsymbol{J}_a \int_0^T \boldsymbol{H}(t)dt \right\|_2$. Knowing that any norm in $\mathbf{R}^{n,n}$ can be rescaled to be submultiplicative (i.e. $||\boldsymbol{AB}||_2 \leq ||\boldsymbol{A}||_2||\boldsymbol{B}||_2$), we obtain

$$|\mu| \leq ||\boldsymbol{J}_a||_2 \left\| \int_0^T \boldsymbol{H}(t)dt \right\|_2$$

which leads to

$$\mathcal{M}\left( \boldsymbol{J}_a \int_0^T \boldsymbol{H}(t)dt \right) \leq ||\boldsymbol{J}_a||_2 \left\| \int_0^T \boldsymbol{H}(t)dt \right\|_2$$

If $L$ is the superior bound of the derivative of the activation function, then, since $\boldsymbol{H}(t)$ is a diagonal matrix, we have $\mathcal{M}\left( \boldsymbol{J}_a \int_0^T \boldsymbol{H}(t)dt \right) \leq LT||\boldsymbol{J}_a||_2$. Therefore, OS-net is stable if $1 < \frac{2}{LT}||\boldsymbol{J}_a||_2^{-1}$ i.e $||\boldsymbol{J}_a||_2 < \frac{2}{LT}$. $\qquad\square$

All in all our minimization problem becomes

$$L(\boldsymbol{x}_o, g(f(\boldsymbol{x}_0))) = ||g(f(\boldsymbol{x}_0)) - \boldsymbol{x}_0||_2^2 \;\; \text{s.t.} \;\; ||\boldsymbol{J}_a||_2 < \frac{2}{LT} \tag{9}$$

and this formulation is equivalent Bach et al. (2011) to

$$L(\boldsymbol{x}_0, g(f(\boldsymbol{x}_0))) = ||g(f(\boldsymbol{x}_0)) - \boldsymbol{x}_0||_2^2 + \alpha||\boldsymbol{J}_a||_2^2 \tag{10}$$

where $\alpha \in \mathbf{R}$ can be fine-tuned using cross-validation. We thus have derived a new regularization strategy that stabilizes the network. By controlling the norm of $\boldsymbol{J}_a = |\boldsymbol{W}_e^T \boldsymbol{W}_d^T|$, we ensure solutions of Equation5 and consequently periodic orbits of Equation2 are stable. We validate these claims in the next section by running a battery of tests on simulated data from dynamical systems known for their chaotic behavior.

## 3 NUMERICAL RESULTS

In this section, we showcace the learning capabilities and stability of OS-net on different regimes of the Rössler Rössler (1976) and of the Sprott systems Sprott (1997). In all of the following experiments, the data was generated using Matlab's ode45 solver. We take snapshots at different time intervals to obtain the data used to train OS-net.

We used the $LBFGS$ optimizer with a learning rate $lr = 1$. and the strong Wolfe Wolfe (1969) line search algorithm for all the experiments. Our code uses Pytorch and all the tests were performed on a single GPU[1].

### 3.1 THE RÖSSLER SYSTEM

As in Bramburger et al. (2021), we consider the Rössler system

$$\dot{x} = -y - z \tag{11}$$
$$\dot{y} = x + 0.1y \tag{12}$$
$$\dot{z} = 0.1 + z(x - c) \tag{13}$$

where $c \in \mathbf{R}$. Rössler introduced this system as an example of simple chaotic system with a single nonlinear term ($zx$). As $c$ increases, this system displays period doubling bifurcations leading to chaotic behavior. Here, we consider the values $c = 6$ and $c = 18$.

### 3.1.1 C = 6, PERIOD-2 ATTRACTOR

First, we set $c = 6$ and initialize the trajectory at $[x_0, y_0, z_0] = [0, -9.1238, 0]$. In this regime, the Rössler system possesses a period-2 attractor Bramburger et al. (2021). We generate the training

Table 1: Norm of $\boldsymbol{J}_a$

| | Part | |
|---|---|---|
| System | Attractor type | $||\boldsymbol{J}_a||$ |
| Rössler, $c = 6$ | Period-2 | 0.9937 |
| Rössler, $c = 18$ | Chaotic | 0.6318 |
| Sprott, $\mu = 2.1$ | Period-2 | 0.0085 |

data by solving the Rössler system using Matlab's ode45 solver with a time step of $0.001$ from $t = 0$ to $t = 10$. We then take snapshots of the simulated data every $50$ step and feed it to OS-net. We build OS-net using the Runge-Kutta 4 (RK4) algorithm with a time step of $0.005$. We chose the snake activation function $x + \frac{1}{0.2} \sin^2(0.2x)$ and set the number of nodes in the hidden layer to be $2 \times 16$. We set $\mu = 0.07$ in Equation 10 and use 10 epochs.
Figure2 (left) shows the training output for the $y$ component. OS-net was able to learn the dynamics accurately by the end of training. The norm of $\boldsymbol{J}_a$ is approximately $0.99$ after training as recorded in Table1. In this case, Inequality1.1 is not strictly enforced but the norm of the matrix $\boldsymbol{J}_a$ is controlled enough so that OS-net renders stable orbits. The elements of the matrix $\Omega$ are concentrated in $[-0.7, 0.7]$.

---

[1]We base our code on the Neural ode implementation in Surtsukov (2019)

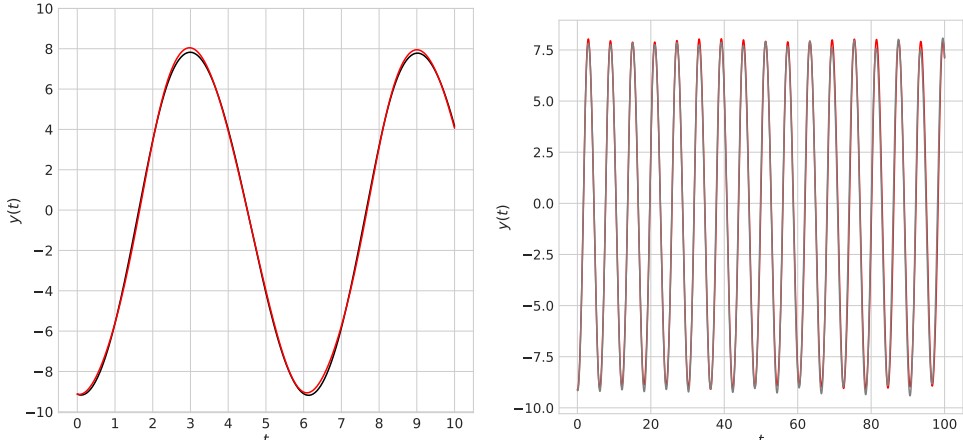

Figure 2: Left: Training data in black along with the learned dynamics in red in the training time interval $[0, 10]$. Right: Target dynamics in gray along with the data generated by OS-net on the time interval $[0, 100]$.

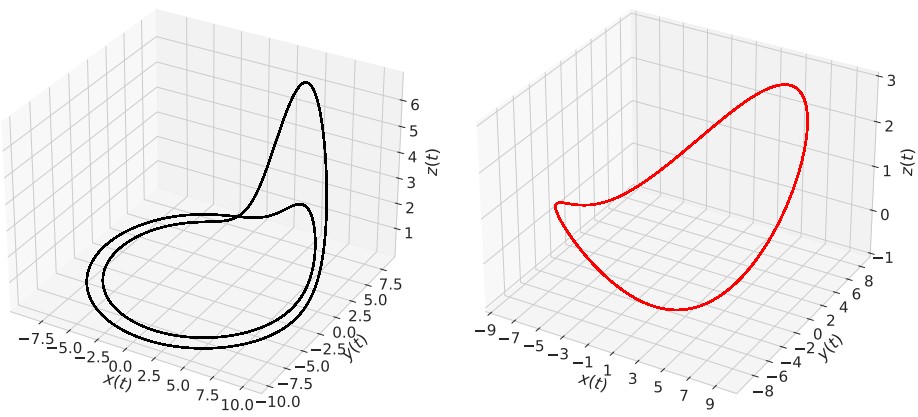

Figure 3: Left: Rössler's period-2 attractor. Right: stable OS-net period-1 attractor (right)

We validate OS-net by propagating a trajectory initialized $[x_0, y_0, z_0] = [0, -9.1238 + 0.01, 0]$ using the learned dynamics. Figure2 (right) shows prediction using OS-net up to $t = 100$ and displays the accuracy of this prediction when compared to the correct dynamics. We assess the stability of OS-net by propagating the trajectory to $t = 10000$. OS-net converges to a stable period-1 attractor while the Rössler system converges to a period-2 one as showcased in Figure3.

**Ablation study:** We compare OS-net with a network obtained by keeping the same architecture but with the regularization in Equation10 switched off. We focus on the Rössler system with $c = 6$ in Equation3.1 and keep the same experiments setup as in section3.1.1. The left side of Figure4 shows that training was successful while the right side shows the dynamics learned by the unregularized network diverge from the true dynamics in the time interval $[0, 10]$. This shows the role of the regularization term in stabilizing the dynamics learned by OS-net. We also compare OS-net to a neural ode with unstructured weights i.e. one where the hidden layer weights $\boldsymbol{W}_e$ and the output layer weights $\boldsymbol{W}_d$ are defined independently. We show the results in Appendix C.

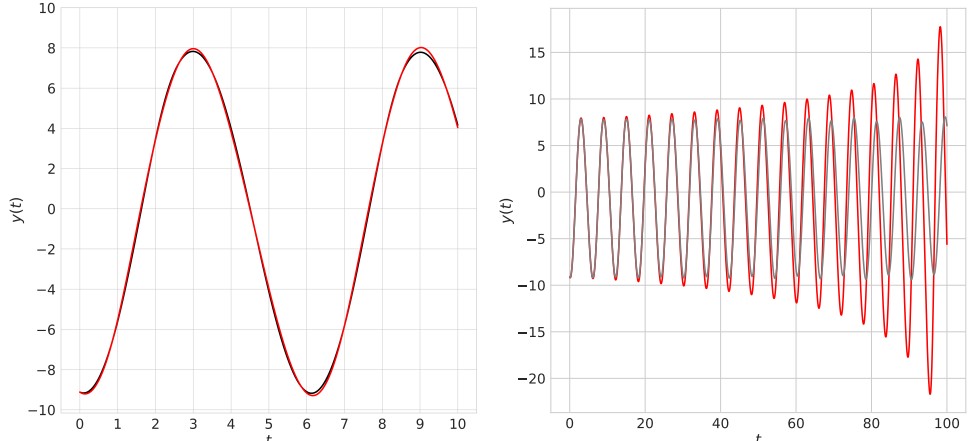

Figure 4: Left: Training data in black along with the learned dynamics in red in the training time interval $[0, 10]$. Right: Target dynamics in gray along with the data generated by OS-net on the time interval $[0, 100]$.

### 3.1.2  C = 18, CHAOTIC BEHAVIOR

We now set $c = 18$ and initialize the trajectory at $[x_0, y_0, z_0] = [0, -22.9049, 0]$. The Rössler system displays a chaotic behavior in this regime. We generate the training data as before but take snapshots every 10 steps. For OS-net, we use RK4 with a step size of $0.005$ and $x + \sin(x)$ as an activation function. The hidden layer size is $2 \times 32$ and the penalty coefficient $\mu = 2$.

Figure5 shows the training output and confirms the ability of OS-net to learn the target dy-

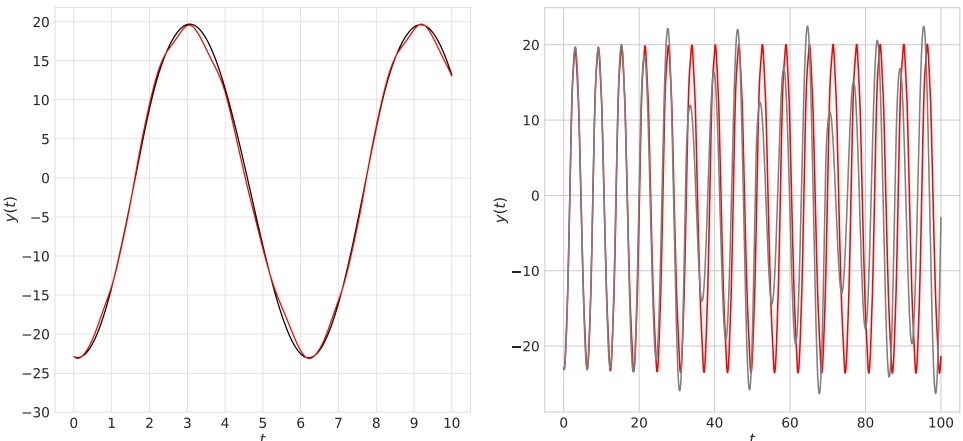

Figure 5: Left: Training data in black along with the learned dynamics in red in the training time interval $[0, 10]$. Right: Target dynamics in gray along with the data generated by OS-net on the time interval $[0, 100]$.

namics. We then use the learned dynamics to generate a trajectory starting at $[x_0, y_0, z_0] = [0, -22.9049 + 0.01, 0]$. Since we are dealing with a chaotic system, the learned dynamics should not be expected to reproduce the training data Bramburger & Kutz (2020). Figure5 shows that OS-net was able to track the chaotic system up to $t \approx 30$. The norm of the matrix $\boldsymbol{J}_a$ was approximately $0.6318$ at the end of training as recorded in Table1. Furthermore, the elements of the matrix $\Omega$ were concatenated between $-0.5$ and $+0.5$. Figure6 displays the chaotic Rössler system and the stable attractor obtained by propagating OS-net's learned dynamics from $t = 0$ to $t = 10000$.

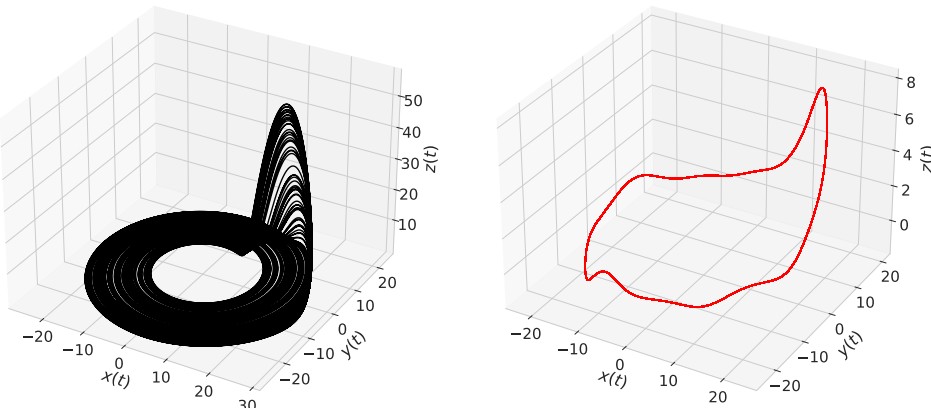

Figure 6: Left: Chaotic Rössler attractor. Right: Stable period-1 OS-net attractor

## 3.2 SIMPLEST QUADRATIC (SPROTT'S) CHAOTIC FLOW

We consider the following system

$$
\begin{aligned}
\dot{x} &= y \\
\dot{y} &= z \\
\dot{z} &= -\nu z - x + y^2
\end{aligned}
\tag{14}
$$

where $\nu \in \mathbf{R}$. This system was introduced in Sprott (1997) and also has period doubling bifurcations as $\nu$ varies. Here we set $\nu = 2.1$ which yields a peiod-2 attractor for Equation14.

We initialize the trajectory at $[x_0, y_0, z_0] = [5.7043, 0.0, -2.12778]$ and solve the system using ode45 on the time interval $[0, 15]$ with a step size of $0.001$. We then take snapshots every 10 step and use the data for training. OS-net is solved using RK4 with a step size of $0.01$ and $x + \frac{1}{0.3} \sin^2(0.3x)$ as an activation function. The hidden layer has $2 \times 16$ nodes and the penalty coefficient $\mu = 1$.

We show in Figure7 (left) the dynamics learned by OS-net for the $y$ component after 20 epochs.

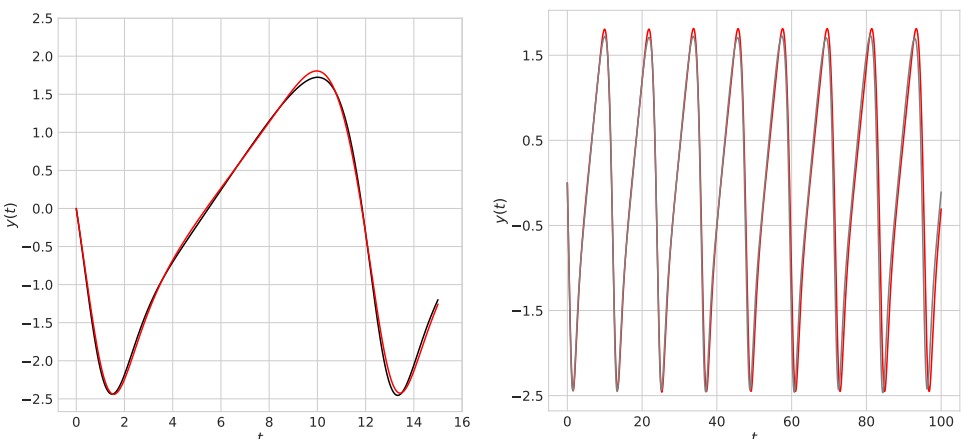

Figure 7: Left: Training data in black along with the learned dynamics in red in the training time interval $[0, 15]$. Right: Target dynamics in gray along with the data generated by OS-net on the time interval $[0, 100]$.

Figure7 (right) also shows how well OS-net tracks the original system in the interval $t = 0$ to $t = 100$. In this case, the norm of the matrix $\boldsymbol{J}_a$ was approximately $8e - 3$ and the elements

of the matrix $\Omega$ are in the interval $[-0.7, \; 0.7]$. Inequality1.1 is strictly enforced here. We then assess the stability of the learned dynamics by generating a trajectory starting at $[x_0, y_0, z_0] = [5.7043 + 0.01, 0.0, -2.12778]$ and evolving it from $t = 0$ to $t = 10000$. Figure8 shows the period-2 attractor of the original system and the stable period-1 OS-net orbit.

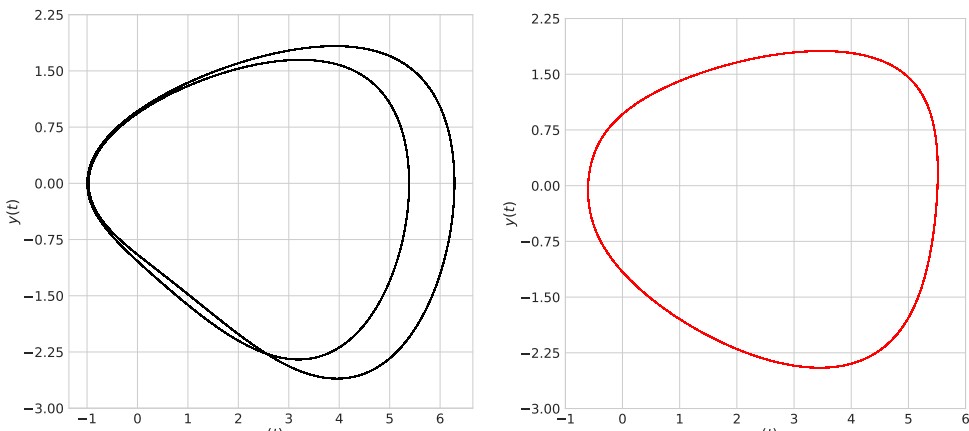

Figure 8: Left: Sprott's period-2 attractor. Right: stable period-1 OS-net attractor

**Note** The current implementation of OS-net uses the adjoint method presented in Chen et al. (2018) which accumulates numerical errors when integrating backward. We circumvent that by using RK4 with a small step size. This results in a computationally expensive implementation that can be improved using the methods proposed in Ott et al. (2021); Zhuang et al. (2020); Zhang & Zhao (2022) that we plan on incorporating into OS-net in the future.

## 4 CONCLUSION

We have presented a new family of stable neural network architectures for periodic dynamical data. The proposed architecture is a particular case of NODES with dynamics represented by a shallow neural network. We leveraged well-grounded ode theory to propose a new regularization scheme that controls the norm of the product of the weight matrices of the network. We have validated our theory by learning the Rössler and Sprott's systems in different regimes including a chaotic one. In all the regimes considered, OS-net was able to track the exact dynamics and converge to a stable period-1 attractor. That indicates that OS-net is a promising network architecture that can handle highly complex dynamical systems. In the future, we aim at explicitly controlling the parameters of the systems of interest by incorporating them into the state vectors that OS-net aims at learning. Additionally, we plan on using OS-net to learn and monitor the orbits of celestial objects that have short orbital periods such as certain exoplanets or three-body systems like Mars-Phobos. This extension of OS-net's applications holds great potential in providing a broader range of stable periodic orbits for the design of spatial missions.

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
