## A   Orbits of dynamical systems

In this section, we recall fundamental results for the stability of periodic orbits of equations of the form

$$\dot{\boldsymbol{x}} = f(\boldsymbol{x}), \ \ \boldsymbol{x}(0) = \boldsymbol{x}_0. \tag{15}$$

where $f \in \mathrm{C}^k(\mathrm{M}, \mathrm{R}^n)$, $k \geq 1$ and M an open subset of $\mathrm{R}^n$.

Chaos theory and the stability of periodic orbits has been widely studied in the literature (Teschl, 2012, Chapter 12), Sprott (2003). They are important concepts in fields like celestial mechanics, biology and chemistry. An important tool to study periodic orbits is the Poincaré or return map. Let $\phi(t, \boldsymbol{x}_0)$ be a periodic solution of Equation 15 with period $T$. Let $I_x$ be the maximal interval where $\phi$ is defined. Denoting

$$\mathrm{W} = \cup_{\boldsymbol{x} \in \mathrm{M}} I_{\boldsymbol{x}} \times \{\boldsymbol{x}\} \subseteq \mathrm{R} \times \mathrm{M},$$

the flow of Equation15 is defined to be the map

$$\Phi : \ \mathrm{W} \to \mathrm{M}, \ \ (t, \boldsymbol{x}) \mapsto \phi(t, \boldsymbol{x}).$$

Let $\gamma(\boldsymbol{x}_0)$ be the associated periodic orbit. The Poincaré map is defined as

$$P_\Sigma(\boldsymbol{y}) = \Phi(\tau(\boldsymbol{y}), \boldsymbol{y}) \tag{16}$$

where $\Sigma$ is a transversal submanifold of codimension one containing one value $\boldsymbol{x}_0$ from the periodic orbit $\gamma(\boldsymbol{x}_0)$, and $\tau \in \mathcal{C}^k(U)$ such that $\tau(\boldsymbol{x}_0) = T$ and $U$ a neighborhood of $\boldsymbol{x}_0$ such that $\forall \ \boldsymbol{y} \in U$, $\Phi(\tau(\boldsymbol{y}), \boldsymbol{y}) \in \Sigma$. It has been proven that the stability of periodic orbits is directly connected to the stability of $\boldsymbol{x}_0$ as a fixed point of the return map $P_\Sigma$. More precisely, we have the following theorem (Teschl, 2012, Chapter 12)

**Theorem 1.** *Suppose $f \in \mathrm{C}^k$ has a periodic orbit $\gamma(\boldsymbol{x}_0)$. If all eigenvalues of the derivative of the Poincare map $DP_\Sigma$ at $\boldsymbol{x}_0$ lie inside the unit circle then the periodic orbit is asymptotically stable.*

It is however generally difficult to compute Poincaré maps and their derivatives explicitly. Fortunately, it was proven in Teschl (2012) that the eigenvalues of the derivative of Poincare map $DP_\Sigma$ at $\boldsymbol{x}_0$ plus the single value 1 coincide with the eigenvalues of the monodromy matrix (see Definition4) of the first variational (FV) equation associated with Equation 15

$$\dot{\boldsymbol{y}} = A(t)\boldsymbol{y}, \ \ \boldsymbol{y}(t_0) = \boldsymbol{I}, \ \ \boldsymbol{A}(t) = d(f(\boldsymbol{x}))_{(\Phi(t,\boldsymbol{x}_0))} \ \text{and} \ \boldsymbol{A}(t+T) = \boldsymbol{A}(t). \tag{17}$$

Therefore, by evaluating the stability of the FV equation associated with a periodic orbit one can assess its stability properties.

## B   Stability of Linear Canonical systems

In addition to the definitions given in the main body of this article, we give more definitions and results that would allow us to prove Theorem 1.

**Definition 4.** *The monodromy $\boldsymbol{U}(T)$ of a periodic linear system $\dot{\boldsymbol{x}} = \boldsymbol{A}(t)\boldsymbol{x}, \ \ \boldsymbol{A}(t+T) = \boldsymbol{A}(t)$ is*

$$\boldsymbol{U}(T) = \boldsymbol{\Pi}(T, t_0)$$

*where $\boldsymbol{\Pi}(t, t_0)$ is the principal matrix solution of the system i.e. $\boldsymbol{\Pi}(t, t_0)$ solves the initial value problem*

$$\dot{\boldsymbol{\Pi}}(t, t_0) = \boldsymbol{A}(t)\boldsymbol{\Pi}(t, t_0), \ \ \boldsymbol{\Pi}(t, t_0) = \boldsymbol{I}.$$

**Definition 5.** *A matrix $\boldsymbol{S}$ is said to be J-unitary if $\boldsymbol{U}^*\boldsymbol{J}\boldsymbol{U} = \boldsymbol{J}$. In particular, the monodromy matrix $\boldsymbol{U}(T, \lambda)$ of Equation6 is J-unitary.*

**Definition 6.** *Let $\boldsymbol{H} \in \mathrm{P}_n(T)$ and consider the boundary value problem (BVP)*

$$\dot{\boldsymbol{y}} = \lambda \boldsymbol{J}_m \boldsymbol{H}(t)\boldsymbol{y}, \ \ \boldsymbol{y}(T) = \boldsymbol{\Xi}\boldsymbol{y}(0)$$

*where $\boldsymbol{\Xi}$ is a J-unitary matrix. The characteristics values of this BVP are the roots (for $\lambda$) of the equation*

$$det(\boldsymbol{U}(T, \lambda) - \boldsymbol{\Xi}) = 0.$$

We now recall the following results from (Krein, 1983, Theorem 6.1 and Theorem 6.2)

**Theorem 2.** *If $\boldsymbol{H} \in \mathrm{P}_n(T)$, then the BVP*

$$\dot{\boldsymbol{y}} = \lambda \boldsymbol{J}_m \boldsymbol{H}(t)\boldsymbol{y}, \quad \boldsymbol{y}(T) = -\boldsymbol{y}(0) \tag{18}$$

*has at least one positive and one negative characteristic value. Furthermore, if we denote $\Lambda_+$ the smallest positive characteristic value and $\Lambda_-$ the largest negative one, then the open interval $(\Lambda_-, \Lambda_+)$ belongs to the central zone of stability of Equation6.*

We recall the following theorem

**Theorem 3** (Krein). *A real $\lambda$ belongs to the central zone of stability of an Equation 6 of positive type, if*

$$|\lambda| < 2\mathcal{M}^{-1}(\boldsymbol{C})$$

*where $\boldsymbol{C} = \boldsymbol{J}_{m_a} \int_0^T \boldsymbol{H}_a(t)$. If $K$ is a matrix, $K_a$ is the matrix obtained by replacing the elements of $K$ by their absolute values.*

*Proof.* Let $\boldsymbol{x}^+(t) = (x_1^+, \cdots, x_n^+)$ be a non trivial solution of Equation5 for $\Lambda_+$ such that $\boldsymbol{x}^+(t + T) = -\boldsymbol{x}^+(T)$. The existence of $\Lambda_+$ and $\boldsymbol{x}^+$ is assured by TheoremB.

The corresponding system is

$$\dot{\boldsymbol{x}^+} = \Lambda_+ \boldsymbol{J}\boldsymbol{H}(t)\boldsymbol{x}^+$$

where $\boldsymbol{x}^+(t + T) = -\boldsymbol{x}^+(t)$ and we set $v_j = \max_{0 \le t \le T} |x_j^+(t)| = |x_j^+(\tau_j)|, \ j = 1 \cdots 2m$.

Let $\boldsymbol{A} = \boldsymbol{J}\boldsymbol{H}(t)$, we have

$$\dot{x^+}_j = \Lambda^+ \sum_k a_{jk}(t)x_k^+, \quad j = 1 \cdots 2m.$$

We integrate these equations from $\tau_j$ to $\tau_j + T$ to obtain

$$-2x_j^+(\tau_j) = \Lambda_+ \sum_k \int_{\tau_j}^{\tau_j+T} c_{jk}(t)x_k^+(t)dt$$

We now take the modulus and obtain

$$2v_j \le \Lambda_+ \sum_k v_k \int_{\tau_j}^{\tau_j+T} |a_{jk}(t)|dt = \Lambda_+ \sum_k c_{jk}v_k$$

Where $\boldsymbol{C} = (c_{ij})_{j=1,\cdots,n} = \boldsymbol{J}_a \int_0^T \boldsymbol{H}_a(t)dt$ and obtain

$$\boldsymbol{v} \le \frac{1}{2}\left(\Lambda_+ \boldsymbol{J}_a \int_0^T \boldsymbol{H}_a(t)dt\right)\boldsymbol{v}$$

where $\boldsymbol{v} = (v_1, \cdots, v_n)$. We now use the following lemma proven in Krein (1983):

**Lemma 3.1.** *If for a nonzero matrix $\boldsymbol{A} = (a_{ij})_{i,j=1,\cdots,n}$ with nonnegative elements there exists a nonzero vector $\boldsymbol{v} = (v_1, \cdots, v_n)$ with nonnegative coordinates such that $\boldsymbol{v} \le \boldsymbol{A}\boldsymbol{v}$, then $\mathcal{M}(\boldsymbol{A}) \ge 1$.*

to state

$$\Lambda_+ \frac{\mathcal{M}\left(\boldsymbol{J}_a \int_0^T \boldsymbol{H}_a(t)dt\right)}{2} \ge 1 \quad \text{i.e.} \quad \Lambda_+ \ge 2\mathcal{M}^{-1}\left(\boldsymbol{J}_a \int_0^T \boldsymbol{H}_a(t)dt\right).$$

In a similar fashion, we obtain $-\Lambda_- \ge 2\mathcal{M}^{-1}\left(\boldsymbol{J}_a \int_0^T \boldsymbol{H}_a(t)dt\right)$. Consequently, $\lambda$ is in the central zone of stability if

$$|\lambda| < 2\mathcal{M}^{-1}\left(\boldsymbol{J}_a \int_0^T \boldsymbol{H}_a(t)dt\right).$$

Hence, if $1 < 2\mathcal{M}^{-1}\left(\boldsymbol{J}_a \int_0^T \boldsymbol{H}_a(t)dt\right)$, then Equation6 is stable. □

## C  ABLATION STUDY (CONT'D)

We compare OS-net to a neural ode with unstructured weights i.e. one where the hidden layer weights $\boldsymbol{W}_e$ and the output layer weights $\boldsymbol{W}_d$ are sampled independently. We first trained this neural ode without regularization which resulted in weights vanishing after one epoch. Adding a regularization term $\mu(||\boldsymbol{W}_e||_2^2 + ||\boldsymbol{W}_d||_2^2)$ where $\mu = 0.07$ as in Section 3.1.1 resulted in successful training as shown in Figure 9. We then propagate the learned dynamics up to $t = 10000$ and see that

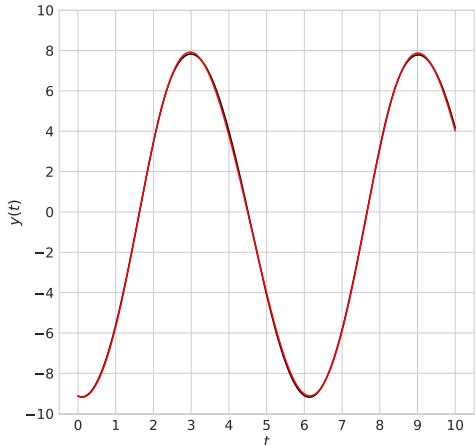

Figure 9: Left: Training data in black along with the learned dynamics in red in the training time interval $[0, 10]$. Right: Target dynamics in gray along with the data generated by OS-net on the time interval $[0, 100]$.

they track the original dynamics well up to $t \approx 100$ before starting to decay and oscillate around $0$ after $t = 1000$ (see Figure 11). We show in Figure 10 the corresponding unstable orbit.

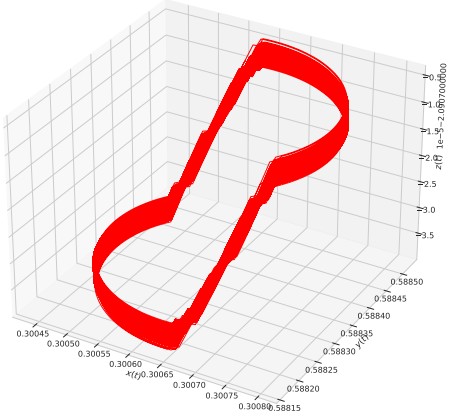

Figure 10: Orbit for the unstructured network.

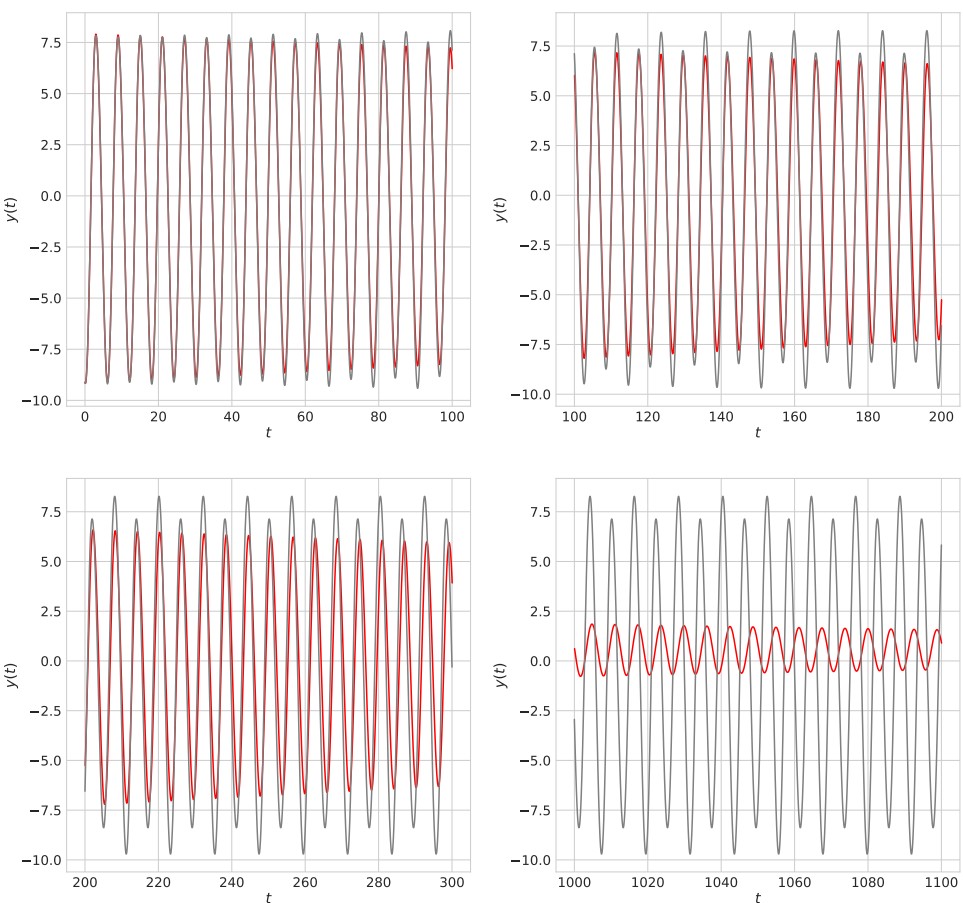

Figure 11: Upper Left: Training data in black along with the learned dynamics in red in the training time interval $[0, 10]$. Upper Right: Target dynamics in gray along with the data generated by OS-net on the time interval $[0, 100]$.