# OpenReview forum: "OS-net: Orbitally Stable Neural Networks"
_ICLR.cc/2024/Conference — Submitted to ICLR 2024_

### Official Review · Reviewer_Yjdz · 2023-10-25

**Soundness:** 2 fair
**Presentation:** 1 poor
**Contribution:** 2 fair
**Rating:** 3
**Confidence:** 2

**Summary:**

This paper introduces a new family of neural network architectures OS-net (Orbitally Stable neural NETworks) for periodic dynamical data. Conditions to ensure stability are derived, and OS-net is shown to work on two systems.

**Strengths:**

There are plenty of figures demonstrating the experimental results.

**Weaknesses:**

1. Although I tried to read through the paper to grasp the main idea, it should be admitted that this paper is quite hard to understand for me. The paper is filled with undefined notations without further explanations, which makes the formulas unreadable and the main idea hard to follow. It would be great if the authors could provide a notation table. Some examples are listed below.

(1) U in Eq. (1).

(2) d in Eq. (4).

(3) M in the second paragraph of section 2.2.

2. There are lots of minor problems and the paper should be checked and organized in a better way to make the paper easier to read.

(1) There are many citations that do not form a part of a sentence but appear in the format style of "Name (year)". The names outside the brackets reduce the readability of the sentence. For example, on the 10th line below Eq. (1), Roy et al. should be put inside the brackets.

(2) On pages 1 and 2, the space between paragraphs is the same as that between lines in the same paragraph.

(3) There are many typos and grammar mistakes that can be easily detected by spelling and grammar software. For example, on the 18th line below Eq. (2), "i.e" should be "i.e.". On the line above Eq. (4), "stabiity" should be "stability". The period at the end of Eq. (4) should be a comma since the sentence is not finished. There is a redundant period on the 5th line in Definition 1.

(4) The section citations are peculiar. In the last paragraph of section 1, "section2" should be "section 2".

3. The learning process (learning dynamics from chaotic systems) and the learning algorithm (OS-net) are not clearly demonstrated. From the aspect of machine learning, it would be better to formulate the learning process by defining the input, output, and target. Meanwhile, it seems that the "chaotic" property does not appear in a formal definition. For the learning algorithm, it would be better to present the algorithm step by step in an algorithm block.

4. There exist minor mathematical errors. In the first paragraph of section 2.2, the authors claim that increasing functions have positive derivatives. This is wrong by considering $f(x) = x^3$. Activation functions used in the paper ($x + \sin x$ and $x + \frac{1}{a} \sin^2 (ax)$) are also increasing but have zero derivatives at some points.

**Questions:**

See weakness.

---

### Official Review · Reviewer_xpia · 2023-10-31

**Soundness:** 2 fair
**Presentation:** 2 fair
**Contribution:** 2 fair
**Rating:** 3
**Confidence:** 2

**Summary:**

The authors propose a method called Orbitally Stable Neural Networks for simulating periodic dynamical data. The main contribution of this paper is the derivation of a new regularization strategy for the constructed neural network to ensure the stability of the dynamic results.

**Strengths:**

S1. The authors propose a novel regularization strategy to achieve stability in predicting dynamic systems.

**Weaknesses:**

First of all, I need to clarify that my review is based on the assumption that the authors have a correct understanding of their references and that their proofs and derivations are correct. I did not carefully check whether their mathematical proofs have any issues.


W1. The introduction of related work is not sufficient. What are the specific works that focus on stability of dynamic systems, and what are the shortcomings of these related works compared to the method proposed in this paper?

W2. The authors claim that they have proposed a family of neural network architectures, but they only proved and experimented with a shallow network featuring a single hidden layer. How can they generalize the proof of the regularization strategy for Matrix J to other forms of neural networks in this family as mentioned in the paper?

W3. I think the explanation of the experimental metrics and the analysis of the results are not detailed enough. For example, how should we evaluate that the results of OS-net shown in the Figures 3, 6, and 8 are better?

W4. I do not agree with the way the authors presented their experimental results. They only present the results of their own method without comparing it to baseline methods to highlight the stability of OS-net. They only used some intuitive images to show the results of dynamic system simulation without relevant metrics, such as MSE, to reflect the numerical difference between the simulation results and the true results.

**Questions:**

Please response W1,W2, W3, W4.

---

### Official Review · Reviewer_2Hj6 · 2023-10-31

**Soundness:** 2 fair
**Presentation:** 1 poor
**Contribution:** 2 fair
**Rating:** 3
**Confidence:** 4

**Summary:**

The paper’s key catch is to use machine learning-based methods, grounded on control theory theoretical results, to stabilise periodic orbits of (typically chaotic) dynamical systems. Specifically, a regularisation strategy focused on controlling the norm of the Jacobian can be derived for a simple architecture consisting of a single hidden layer plus a linear output layer. Authors show the effectiveness of their method on two benchmark tasks: the Rossler and Sprott’s systems.

**Strengths:**

The idea of learning Neural ODEs to stabilise periodic orbits seems fairly original.

The quality of the theoretical section is generally good and quite clear.

I find particularly interesting the links established between known results of the stability theory of dynamical systems and ML applications of Neural ODEs.

**Weaknesses:**

The link with related works and literature is poor. I struggled to understand whether the goal of the paper is the same of [1]. In [1] it is clearly stated that the goal is to detect unstable periodic orbits of a dynamical system. From figure 2, I deduce that the training data is the unstable 1-period solution of the Rossler for c=6. Thus, the OS-net is using the unstable 1-period solution itself to learn an ODE that has stable dynamics converging to such 1-period solution. If that is right, then the contribution of the paper looks poor because it is detecting an already detected orbit. I strongly suggest the authors to state in a much clearer way what is the goal, and the detailed experimental procedure.

Therefore, I’m led to assume that the contribution of the paper is to stably track unstable solutions present in the chaotic regime, as for the case of Rossler with c=18. However, if the training data was from t=0 to t=10, then it is not fair to claim that the OS-net learned the periodic solution from the chaotic trajectory, because the trajectory starts to deviate from the periodic motion only for t>30. I guess the OS-net converges to this periodic solution just because the transient of the trajectory used as training data was looking periodic. Would the OS-net work well in reproducing an 8-period solution of the Rossler in the chaotic regime? If the system quickly starts to wander erratically, then there is not enough time in the training data to explore the 8-period solution, then what the OS-net would learn?

Overall, the experimental setting seems to reduce to the learning of an ODE whose dynamics converge to a periodic solution that coincides with the trajectory provided as input. In conclusion, the experiments are not well devised to evaluate the effectiveness of the proposed methodology, and so the contribution and impact of the paper is blurred.

[1] Bramburger, Jason J., J. Nathan Kutz, and Steven L. Brunton. "Data-driven stabilization of periodic orbits." IEEE Access 9 (2021): 43504-43521.

**Questions:**

Q1) What is precisely the problem to solve with the proposed methodology?

Q2) Authors show that their regularisation method is necessary for stability. It would have been nice to see a similar experiment for the activation function. Is it really necessary to have these peculiar activation functions?

Q3) How can we compare this methodology with the one proposed in [1]? Any measure to compare the two? Some ideas for measures: computational time to train the model, speed of convergence to the solution, NRMSE, number of unstable periodic orbits NOT detected in a pool of well-known chaotic dynamical systems.

MINORS:
* First line of pag 3, J should be bold.
* At the end of definition 2, what integral of A(t) means shouldn’t depend on H(t).
* In theorem 1, clearly define the inverse of M calligraphic.

---

### Official Review · Reviewer_YZQH · 2023-10-31

**Soundness:** 2 fair
**Presentation:** 2 fair
**Contribution:** 3 good
**Rating:** 5
**Confidence:** 2

**Summary:**

The paper introduces a novel family of neural network architectures tailored for periodic dynamical data. By leveraging periodic ODE theory, the authors derive conditions on the network weights to ensure the stability of the resulting dynamics. The primary application of OS-net is to elucidate the dynamics of the Rössler and Sprott’s systems, renowned for their period-doubling attractors and chaotic behavior. The paper provides an in-depth exploration of the stability of periodic orbits in nonlinear dynamics and its application to neural networks. Additionally, the authors underscore the significance of the activation function in maintaining periodicity and propose two activation functions optimized for periodic data.

**Strengths:**

By introducing a regularizer, the paper introduces a novel architecture, OS-net, that bridges the gap between neural networks and periodic dynamical systems, offering a fresh perspective on the design of neural networks for specific types of data.

**Weaknesses:**

1. The authors highlight the pivotal role of the activation function, yet the rationale behind the specific activation functions chosen, especially $x+sin(x)$ leading to a periodic system, could be further expounded upon.
2.  While the final optimization equation (10) is derived from (9), its original form (5) might not necessarily solve (6). It remains unclear how the authors ensure that the final solution adheres to eq(10).
3. $g$ in equation (9) is not defined.
4. The paper lacks a discussion on computation time.

**Questions:**

Minors:

1. Citation style: Please use \texttt{citep} instead of \texttt{cite}.
2. There are typos in Definition 1.

---

### Meta-Review · Area_Chair_pThK · 2023-12-03

**Metareview:**

This paper introduces new architectures for modelling dynamical systems, and presents experiments over classic dynamical systems benchmarks.

The reviewers unanimously gave scores tending rejection (5, 3, 3, 3).

The reviews could be characterized by two categories:
* The reviewers who gave 3's were underconfident about their familiarity with the topic at hand, and felt that the paper lacked proper introduction and importance to the topic, and also lacked standard items in a paper (baselines, references to previous work, experimental ablations, etc.)
* Reviewer 2Hj6 gave a score of 5, with more familiarity with the topic, but raised concerns with the experimental procedure of the paper.

Thus, I recommend rejection.

**Justification For Why Not Higher Score:**

Despite some of the reviewers' lack of confidence, the scores were quite negative (5, 3, 3, 3), leading to clear reject.

**Justification For Why Not Lower Score:**

N/A

---

### Decision · Program_Chairs · 2024-01-16

Reject